

# Design and analysis of management platform based on financial big data

Yuhua Chen[1,2], Hasri Mustafa[2], Xuandong Zhang[3] and Jing Liu[1]

[1] Zhongyuan Institute of Science and Technology, Zhengzhou, China
[2] University Putra Malaysia, Serdang, Malaysia
[3] Silla University, Busan, South Korea

## ABSTRACT

Traditional financial accounting will become limited by new technologies which are unable to meet the market development. In order to make financial big data generate business value and improve the information application level of financial management, aiming at the high error rate of current financial data classification system, this article adopts the fuzzy clustering algorithm to classify financial data automatically, and adopts the local outlier factor algorithm with neighborhood relation (NLOF) to detect abnormal data. In addition, a financial data management platform based on distributed Hadoop architecture is designed, which combines MapReduce framework with the fuzzy clustering algorithm and the local outlier factor (LOF) algorithm, and uses MapReduce to operate in parallel with the two algorithms, thus improving the performance of the algorithm and the accuracy of the algorithm, and helping to improve the operational efficiency of enterprise financial data processing. The comparative experimental results show that the proposed platform can achieve the best the running efficiency and the accuracy of financial data classification compared with other methods, which illustrate the effectiveness and superiority of the proposed platform.

# INTRODUCTION

Traditional financial accounting will become limited by new technologies which are unable to meet the market development. It is urgent to make reforms and innovations. It is an unchangeable trend to integrate artificial intelligence into the financial management and big data development of enterprises. It can simplify the traditional accounting work with high efficiency, so as to meet the rapid and high-quality development of enterprises (*Königstorfer & Thalmann, 2020*). Therefore, the development of artificial intelligence technology will further promote financial management, and the staff will constantly enhance their professional ability and level, which will also bring new vitality to the development of enterprises and adapt to the changes and development of the new era.

Because of the large scale and weak regularity of financial data, it is necessary to optimize the classification of financial data. Automatic classification of financial data combined with business processes is conducive to improving the ability of classification management and information analysis of financial data. With the rapid development of big data and

Corresponding author
Yuhua Chen,
yuhuachen2021@gmail.com

information technology, the automatic classification technology of financial data has made good development (*Yue et al., 2021*). Through feature extraction and fusion clustering of financial data, feature information of internal association rules of financial data is extracted, and then financial data is automatically classified and identified according to the distribution of feature information, which can improve the business process management ability of financial data (*Pejić Bach, Krstić & Seljan, 2019*). Anomaly data usually has the characteristics of outlier, deviation and isolation, and the traditional anomaly detection algorithm relies on the computing performance of a single computer to identify it. The recognition speed of this method is slow, and the real-time performance of enterprise financial data is usually high, so Hadoop parallel computing has an important application value. The Hadoop platform, as one of many distributed storage methods, not only has the ability to share resources, but also can better solve the problems in the storage process. Hadoop can not only solve the distributed storage problem, but also provide a more comprehensive solution, with better data storage performance, fault tolerance, capacity expansion, reading speed and so on (*Qureshi & Gupta, 2014*). MapReduce framework in Hadoop can efficiently read and utilize data and speed up data processing, which mainly works in parallel computing (*Rajendran, Khalaf & Alotaibi, 2021*). Hadoop has good capacity expansion performance, and at the same time, it has good performance expansion and reliability. To cope with heterogeneous environment, it is necessary to design the architecture of MapReduce is designed to be very complex, but it brings difficulties to deployment, maintenance and management. Moreover, dynamic node joining or exiting in a Hadoop cluster may result in unbalanced load distribution in the cluster, slow system response, and slow service processing.

To solve the above problems, this article classifies financial data automatically by the fuzzy clustering algorithm, and detects abnormal data by LOF algorithm with neighborhood relation. In addition, a financial data management platform based on distributed architecture is designed, which combines MapReduce framework with the fuzzy clustering algorithm and the LOF algorithm, and uses MapReduce to operate in parallel with the two algorithms, thus improving the accuracy of the algorithms, and further improving the operational efficiency of enterprise financial data processing.

## LITERATURE REVIEW

### Financial data processing technology

At present, the processing technology of financial big data is very extensive. In terms of data classification, *Zhang (2018)* proposed a design method of financial data classification and statistics system based on cloud computing technology. The retrieval structure model of financial database was constructed by using multiple regression analysis method. The statistical analysis method was used for automatic statistics of financial data to achieve optimal classification of financial data. However, the accuracy of this method for large-scale financial data classification processing is not high; *Xiong et al. (2018)* proposed a classification and statistics method of financial data based on decision tree classification. The empirical mode decomposition method was used for automatic classification and

identification of financial data. The adaptive fusion processing of financial data was realized, and the fuzzy clustering ability of financial data was improved. The software design of automatic classification system for financial data was carried out in embedded environment, but the adaptive control performance of the system is not good in the classification of financial data of different business processes.

In the aspect of abnormal data detection, the existing anomaly detection algorithms usually only give the binary attribute of whether a point is abnormal or not. However, the data set in the enterprise's financial situation is complex and diverse, and it cannot be simply represented by Boolean variable. For this reason, *Papadimitriou, Kitagawa & Gibbons (2003)* proposed a density based local anomaly factor detection algorithm, which can be used to determine the abnormal degree of an object. This method has good effect and strong adaptability, but it has the shortage of large amount of calculation, and does not consider the weight of different attributes when calculating the distance between objects, and does not reflect the different contributions of different attributes to outliers. *Karale et al. (2021)* put forward the local correlation integral (LOCI), and introduced the multi-granularity deviation factor (MDEF) to measure the degree of abnormal objects. By comparing the number of objects contained in the k-neighborhood of an object with the average number of objects in the k-neighborhood of all objects in its neighborhood, which does not need to calculate the density of points directly. *Nowak-Brzezińska & Horyń (2020)* proposed the connectivity based outlier factor (COF), where the neighborhood is determined according to the given minimum number of k-neighbors and the connectivity of the data object, and the average connection distance with its neighborhood is calculated, and the average connection distance ratio is used as the COF. To some extent, the above algorithms solve the shortcomings of large computation amount of LOF, and the operation efficiency is higher than that of the LOF algorithm. However, the overall computation amount is still relatively large, and the operation efficiency is not high enough when dealing with large-scale data. At the same time, it does not reflect that different attributes have different contributions to outliers.

## Hadoop technology

After years of development, the research on Hadoop in the financial field has achieved fruitful results (*Sun, Wang & Yuan, 2021*; *Li, Zhao & Chenguang, 2015*). For example, Hadoop technology is used to deal with the storage and query of massive historical data in the Agricultural Bank of China. In terms of data processing, it attempts to store massive relational data in the HBase database according to the characteristics of large amount of relational data. In the implementation process of mutual management system, the data storage mode in Hadoop technology is adjusted according to the actual business requirements. The relational database and nonrelational database are used to store the data with different correlation degree. In addition, many researchers have studied the improvement of distributed file system, the optimization of scheduling algorithm and the improvement of the MapReduce framework.

From the optimization and improvement of the Hadoop Distributed File System (HDFS) (*Qin et al., 2012*): first, the original architecture of HDFS is improved, and the system is

designed to be more lightweight, so as to reduce the bottleneck problem of NameNode in the original architecture (*Raju, Amudhavel & Pavithra, 2014*). From the improvement and optimization of the scheduling algorithm application, it can be seen that the first is the improvement of Hadoop's inference mechanism, which enables nodes with strong processing performance to help poor performance nodes complete tasks and improve system performance; the second is to improve the data location. MapReduce is divided into two steps: one is the mapping map stage, the other is the reduction stage. After the end of the map phase, the data needs to be transferred to the reduce stage, which will occupy a large number of network channels; The third is the improvement of the shuffle stage. In the whole process of data processing, the shuffle stage takes the longest time. Therefore, how to shorten the running time of the shuffle stage has become the focus of many scholars. Fourth, in addition to the above three improved methods, some scholars have proposed scheduling methods based on data budget judgment and pre allocation (*Wang, 2018*). Moreover, from the improvement of the MapReduce framework, the first is to optimize the performance of tasks executed in the MapReduce system; the second is to optimize the performance reduction of MapReduce framework due to excessive dependence on disk for data result transfer; the third is to optimize the execution efficiency of iterative computing tasks under the existing execution framework.

# DESIGN OF FINANCIAL DATA MANAGEMENT PLATFORM BASED ON HADOOP DISTRIBUTED ARCHITECTURE

## MapReduce framework

In this article, the MapReduce framework is combined with the fuzzy clustering algorithm and the NLOF algorithm, and MapReduce is used for parallel operation with the two algorithms, so as to improve the performance of the algorithm as well as the accuracy of the algorithm. The overall structure of the cluster is shown in Fig. 1.

The model consists of four layers: fuzzy clustering of financial data, Hadoop parallel computing cluster, abnormal data detection module and user control terminal. The architecture of Hadoop computing cluster is completely distributed, where the computing cluster can provide users with a certain number of data storage functions. The classification model is judged by observing the initial samples, and the automatic classification of financial data is realized by segmented detection method. Moreover, the data detection module is constructed with the improved NLOF algorithm, and the final output is the outlier data set. The user control terminal is the upper computer part, which can view the abnormal data through the UI interface and control the computer cluster.

## Virtual resource management

In the distributed architecture, the financial data management platform mainly includes data acquisition equipment, virtual machine and service cluster. Combined with virtualization technology, the service cluster part is virtualized into a computing data
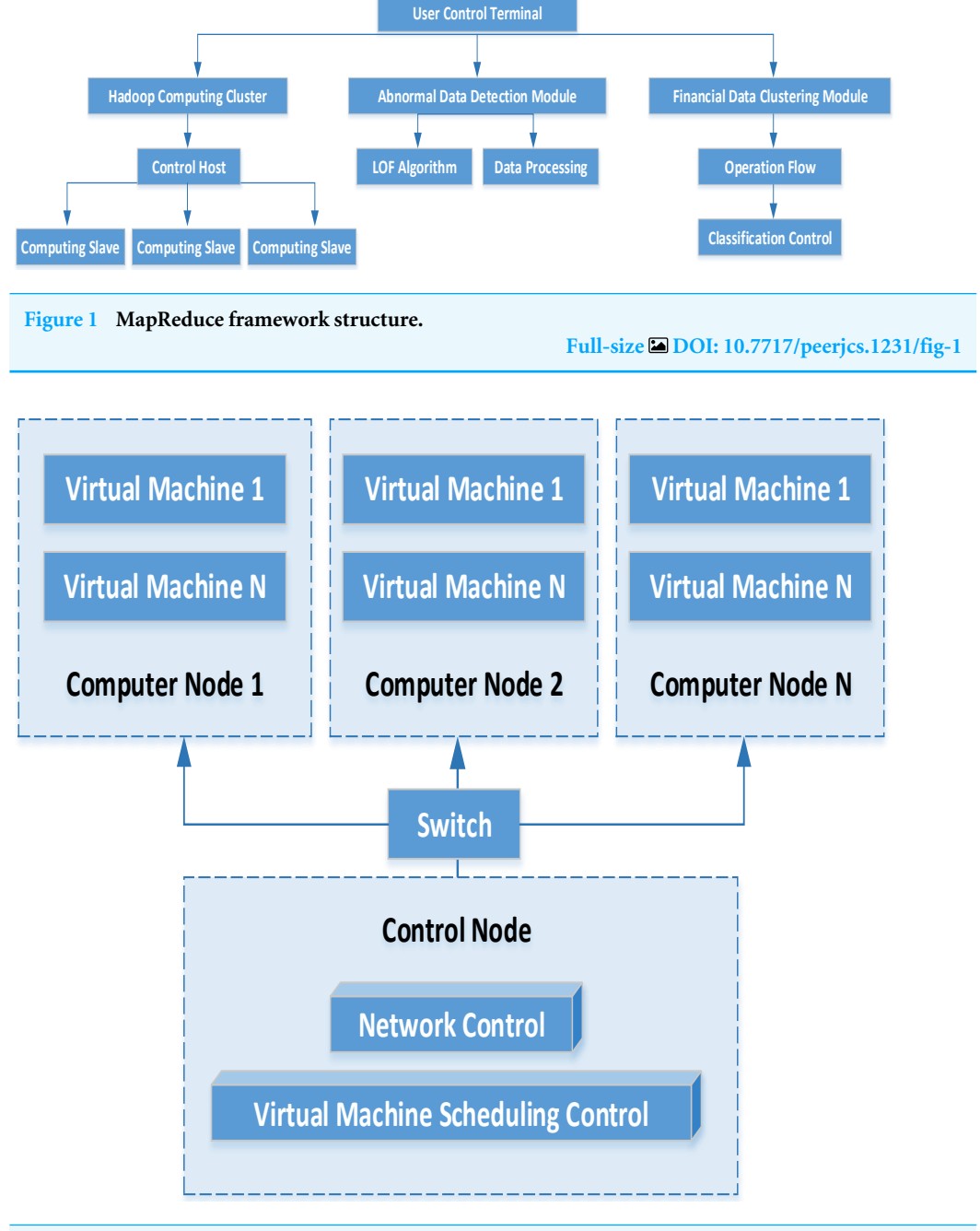

**Figure 1** MapReduce framework structure.

**Figure 2** Operation process of virtual machine.

resource pool. Finally, the flexible call of various resources is realized. The specific flow of virtual machine operation and management is shown in Fig. 2.

Through the analysis of the virtual machine operation management model, we can find that the virtual machine instance and the corresponding hardware configuration are automatically deployed with the support of cloud platform. Assuming that under multiple

computing nodes, the virtual machine operation will establish the corresponding running snapshot and save the running state of the virtual machine. When the virtual machine needs to be started again in the future, the corresponding virtual machine snapshot can be selected to complete the deployment task again. Through the distributed storage service, the data information generated by the virtual machine can be saved on the storage service. While when the virtual machine starts again unexpectedly, it can connect the existing storage service and continue to run under the existing information. The virtual machine can manage and transfer the instructions from the virtual machine to the virtual control platform.

## Cluster management

Web application clusters mainly provide session replication. Through session replication, support for JSP and servlet cluster is realized. Cluster server can complete file copy or database copy by connecting with different types of servers, which are two ways of session replication. When the fault occurs, other available members in the cluster will receive subsequent client requests, read the cookie or write the session ID in the request URL to obtain the relevant status and continue to process the client transaction. Memory replication is realized by restoring session state in backup server, and memory session state in backup server is sent by primary server.

The business cluster is one of the ways to realize the business application cluster, in which requests are distributed across servers according to load balancing algorithm. When the target service and stub programs exist on the same server at the same time, load balancing is not performed. At the same time, whether to re-execute the call of the remote stub business method on the cluster service instance and the routing mode of the home skeleton call in the cluster can also be determined by the fault recovery service.

# FINANCIAL DATA PROCESSING ALGORITHM

## Classification of financial data based on fuzzy clustering

On the basis of data fusion processing of collected financial data by distributed cloud computing technology, the algorithm of automatic classification of financial data is optimized, and a classification model of financial data based on business process is proposed.

### Problem description

The grouping sample test analysis method is used to analyze the correlation of financial data, and the test set is associated with the category characteristics of business process information. The space–time norm of automatic classification of financial data satisfies the following boundary conditions:

$$\begin{cases} f : R^n \times R^n \times S \\ g : R^n \times R^n \times S \rightarrow R^{n \times d} \\ h : R^n \times R^n \times S \end{cases} \tag{1}$$

Taking the fuzzy cluster distribution of business process as the central vector, the multivariate discriminant model is used to predict the number of financial data categories,

and the statistical function of financial data classification is obtained as follows:

$$h : R^n \times R^n \times S$$
$$\delta(t) : [0, T] \to R \qquad (2)$$
$$v(dt, du) = v(dt, du) - \pi(du)dt$$

According to the correlation analysis results of financial data, the quality of financial data classification is evaluated, and the combination problem of financial data classification is described as follows:

(1) $C([a, b], R)$ is the classification continuous function of financial data of $[a, b]$ in classification space R, and the cross-term samples of classification attributes meet the following requirements:

$$-\tau < \delta(t) < t, |\delta(t) - \delta(s)| < p|t - s| \qquad (3)$$

(2) In the initial clustering space, the information fusion center $C([a, b], R)$ of financial data classification has upper and lower boundaries, and (Eq. 4) should be satisfied

$$E|\zeta(t) - \zeta(s)|^2 < K(t - s)^2, -\tau < s < t < 0. \qquad (4)$$

### Data classification

Combined with Bayesian judgment method, automatic classification and classification robustness analysis of financial data are realized. Classification template of financial data under business process satisfies positive multi-solutions. $f : \to R$, internal control index $\alpha > 0$, the linear discriminant function of classification statistics is:

$$D_0^\alpha f(t) = \frac{1}{\Gamma(n - \alpha)} \left( \frac{d}{dt} \right)^n \int_0^t \frac{1}{(t - s)^{\alpha - n + 1}} f(s) ds. \qquad (5)$$

By using validity test and model fitness test, the characteristic functional of data automatic classification is obtained:

$$\frac{1}{m} \sum_{i=0}^{p} a_i m^i \sum_{k=-q/2}^{q/2} b_k c_k^i = a_1. \qquad (6)$$

Under fuzzy constraints $c_k = -c_{-k}$, if $q = 4, b_2 = b_{-2} = 1, b_1 = b_{-1} = 2, b_0 = 0$, Cauchy-Had-Amard differential equation is used for statistical analysis of financial business process data, and (Eq. 7) can be obtained.

$$u(t) = c_1 t^{\alpha - 1} + c_2 t^{\alpha - 2} + \ldots, + c_N t^{\alpha - N}. \qquad (7)$$

The distributed adjacent characteristic quantities of financial business process data are calculated. In the statistical data area, the classification closure function on $IR^d$ is denoted as:

$$\hat{f}(\xi) = (2\pi)^{-d/2} \int_{IR^d} e^{-ix \cdot \xi} f(x) dx \qquad (8)$$

if $s \geqslant 0$, its stationary period satisfies (Eq. 9):

$$
\begin{aligned}
\dot{H}_x^s(IR^d) \quad &= \left\{ f : \|f\|_{\dot{H}_x'(IR')} := \||\nabla|^s f\|_{L_x^2(IR^d)} \right. \\
&= \left. \||\xi|^s \hat{f}\|_{L_2^2(IR^d)} < \infty \right\}.
\end{aligned}
\tag{9}
$$

After iterative selection, taking the business process as the identification parameter, the final discriminant function of automatic classification of financial data is obtained as follows:

$$
\|f\|_{L_i^q Z_1'(I \times IR^k)} = \left( \int_I \left( \int_{IR} |f(t,x)|^r \, dx \right)^{q/r} dt \right)^{1/q}.
\tag{10}
$$

By observing the initial sample, the classification model is judged, and the automatic classification of financial data is realized by the subsection detection method, and the improved design of classification algorithm is realized.

## Abnormal data detection based on improved LOF algorithm

The traditional LOF algorithm calculates the outlier factor of each element in the data set, and then judges whether the element point is an outlier. The calculation of feature points of elements is too simple, so some scholars have added neighborhood relation to LOF algorithm. Therefore, the calculation method of outlier factor is more accurate, and the most remarkable feature of the algorithm is to avoid the possible misjudgment of edges when multiple elements are too close.

Let the element in dataset d is $o_i$ andthe other element is $p_i$, then the distance between objects $o_i$ and $p_i$ is:

$$
d(p,o) = \sqrt{\sum_{i=1}^{d} (p_i - o_i)^2}.
\tag{11}
$$

If the relationship between k neighborhood and $d(p,o)$ is equal, the following conditions must be satisfied:

(1) There are at least k element, $h \in D$, which satisfies $d(p,h) \leqslant d(p,o)$;

(2) There are at most $k-1$ element, $h \in D$, which satisfies $d(p,h) > d(p,o)$.

Then the k-neighborhood distance of p is defined as: $N_{k-distance}(P) = Q|P \in D \wedge d(P,Q) \leqslant k - distance(P)$, where Q is the k-neighborhood element points of P. Then the inverse k-neighborhood of P is defined as follows:

$$
RNN_k(p) = \left\{ q|q \in D, p \in NN_k(q) \right\}
\tag{12}
$$

(3) For any point P, its attainable precision, that is, the inverse of the k-neighborhood distance object of P is:

$$
den(p) = \frac{1}{k - dis(p)}.
\tag{13}
$$

According to (Eq. 13), the outlier factor of point P in the improved LOF algorithm can be defined as:

$$
INFLO_k(p) = \frac{den_{avg}(IS_k(p))}{den(p)}.
\tag{14}
$$

In (Eq. 14), den and $IS_k$ can be defined as:

$$\text{den}_{\text{avg}}\left(\text{IS}_k(p)\right) = \frac{\sum_{o \in I_S(p)} \text{den}(o)}{\left|\text{IS}_k(p)\right|}$$

$$\text{IS}_k(p) = \text{NN}_k(p) \cup \text{RNN}_k(p) \tag{15}$$

It can be seen that the LOF algorithm with neighborhood relation considers both positive and negative k-neighborhood distances, which can effectively represent the density characteristics of data center points. It can be seen from the above definition that if the value of the selected data fixed point P is 1, which means that the data density of the point P is close to the data points in its field; If the value of the selected data fixed point P is not equal to 1, it means that P is in an area with low density or high density, and the fixed point P is an abnormal data point. In order to further improve the selection ability of data attributes, when calculating the distance of k-neighborhood, the weighting algorithm is used to determine the information entropy of data attribute features.

Assuming X is the value of a random variable, and its data set of random variables is defined as S(X), then the data entropy of X is:

$$E(X) = -\sum_{x \in S(X)} P(x) \log[P(x)]. \tag{16}$$

The premise of calculating weighted distance is to define the outlier attribute of data, which can be defined as:

$$w_i = \frac{\sum_i E(A_i)}{d}. \tag{17}$$

The weighted distance of the data can be defined as:

$$d(p, q, w) = \sum_{i=1}^{d} w_i \left(f_{A_i}(p) - f_{A_i}(q)\right)^2. \tag{18}$$

If the data is used to characterize the characteristics of outliers, the weight value will be greater than 1; If the data does not belong to the characteristics of outliers, the weight value is 1, which has no influence on the whole data.

## EXPERIMENT AND ANALYSIS

### Classification effect of financial data

The experiment is based on the MATLAB platform, and the classification attribute of financial data is 12. Under the guidance of business process, the sampling scale of statistical information of financial data is 1,000 MBit, and the simulated data set includes two partitions with the size of 36 MB. To verify the effectiveness of this method in classifying financial data, the following comparative experiments are designed. Taking the methods adopted in *Zhang (2018)* and *Xiong et al. (2018)* as a comparison scheme, the accuracy of financial data classification by different models is tested, and the comparison results are shown in Fig. 3.

Analysis of Fig. 3 shows that with the continuous increase of the number of experimental iterations, the accuracy of different systems is constantly changing, showing a downward

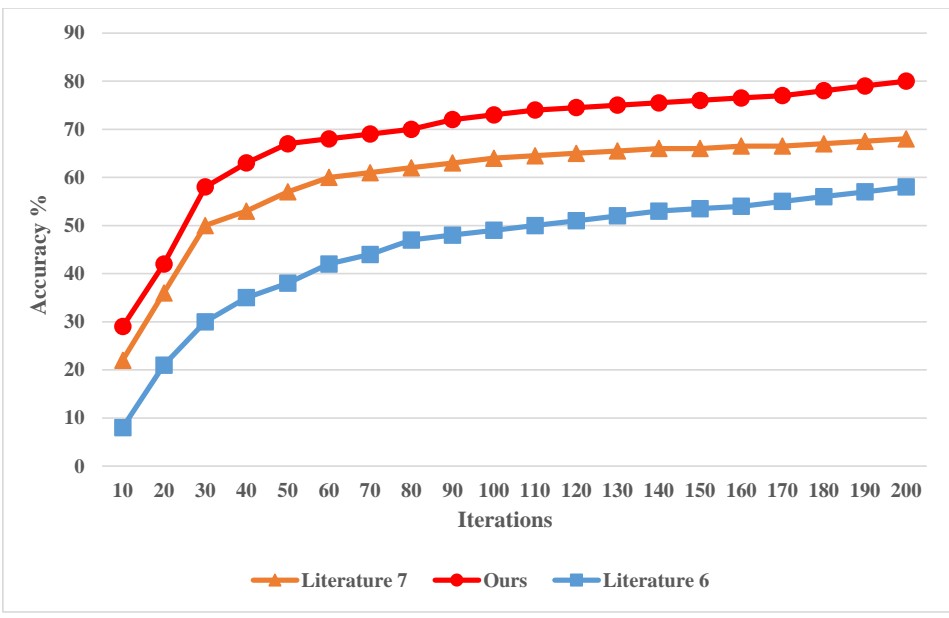

**Figure 3** Accuracy of financial data classification of different models.

trend. Among them, the accuracy of the method in *Zhang (2018)* decreases the most, and its accuracy is also lower than the method in this article and the method in *Xiong et al. (2018)*. In addition, the accuracy decline of the method in this article is similar to that of the method in *Xiong et al. (2018)*, but the accuracy of the system in this article is higher, which proves that the misclassification rate of the financial data classification system in this article is low and can effectively improve the statistical analysis ability of financial data.

## Abnormal data detection results

This experiment will analyze and verify the accuracy and effectiveness of the target algorithm, so the data set selected in this article is the KDD-CUP1999 data set with similar characteristics to financial data. Figure 4 shows the experimental results of the LOF algorithm and the NLOF algorithm on the KDD-CUP1999 data set. In the experiment, the accuracy of adding mesh reduction and information entropy to the algorithm for attribute weighting is compared.

As can be seen from Table 1, grid number reduction has no influence on the accuracy of the model. After introducing information entropy to give different weights to attributes, the accuracy on the KDD-CUP1999 data set is improved. The accuracy of NLOF is 89%, that of NLOF (without information entropy) is 84%, and that of the LOF algorithm is 84%. The former two are the same, but the latter two are the same, and the former two are better than the latter two.

Figure 4 shows the running efficiency comparison of the LOF algorithm and the NLOF algorithm on single machine and cluster respectively.

As can be seen from Fig. 4, at the beginning, the running path of NLOF algorithm on a single machine is not as efficient as the LOF algorithm, which is mainly because NLOF
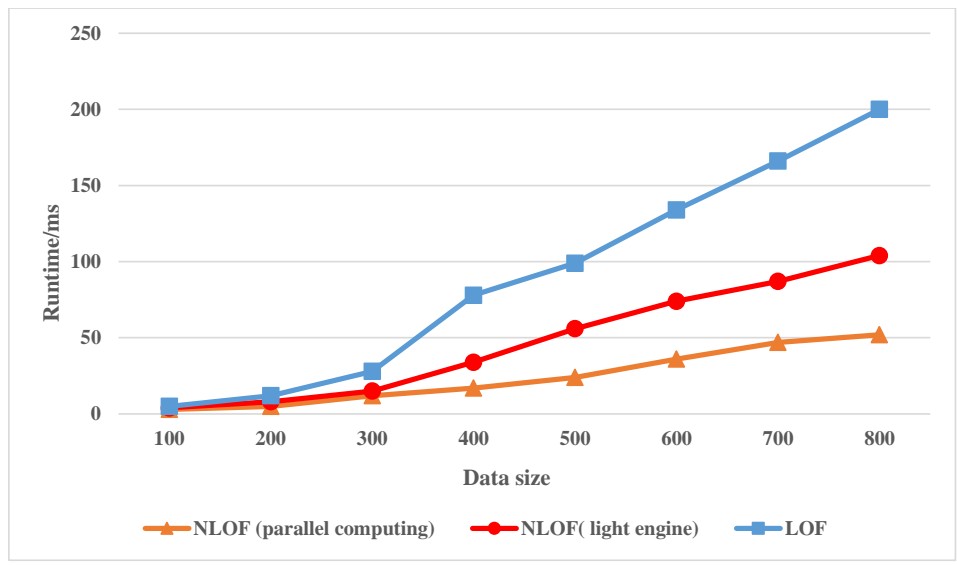

**Figure 4** Running efficiency comparison of the NLOF algorithm.

**Table 1** Comparison of accuracy of the NLOF algorithm.

| | Accuracy/% |
| --- | --- |
| NLOF | 89 |
| NLOF (without mesh reduction ) | 89 |
| NLOF (without information entropy) | 84 |
| LOF | 84 |

algorithm needs to divide the information entropy difference to calculate the weight of each attribute. With the growth of data, it gradually shows that NLOF algorithm is more efficient than the LOF algorithm, it mainly because the NLOF algorithm does not calculate the LOF value of all data sets, but first reduces the data with grid, eliminates a large number of non-abnormal points in high-density areas, and only needs to calculate the LOF value of data in sparse areas, which greatly reduces the calculation amount of the NLOF algorithm.

Then we compare the running efficiency of NLOF algorithm on a single machine with that on a cluster. As can be seen from Fig. 4, at the beginning, the efficiency of the NLOF algorithm on the cluster is not as high as that of the NLOF algorithm on a single machine. This is mainly because when the parallel NLOF algorithm runs on the cluster, it takes a certain amount of time for each node to start and initialize its tasks. When the amount of data is not large, its efficiency is not obvious. However, when the amount of experimental data is large, its efficiency is obviously higher than that of running on a single machine. With the increase of data, the efficiency of the parallel NLOF algorithm becomes more and more obvious.

## CONCLUSION

Using the parallel algorithm to analyze financial big data can effectively improve the speed and accuracy of algorithm identification. Based on the distributed Hadoop architecture, In this article, MapReduce, the fuzzy clustering algorithm and NLOF are used for parallel operation, which helps to improve the accuracy of the algorithm in calculating outliers, and effectively avoids the probability of misjudgment at the edge of elements when multiple elements are too close. In addition, the attribute classification of financial data are combined with the business process, where the fuzzy cluster distribution of business process is taken as the central vector, and the automatic classification of financial data is realized by the segmented detection method, thus the improved design of classification algorithm is realized. The results show that the training performance of the proposed algorithm on the KDD-CUP1999 data set is good, and when the experimental data is large, its efficiency is obviously higher than its running efficiency on a single machine. The financial data management platform based on distributed Hadoop architecture proposed in this article can improve an enterprise's business management and analysis ability of financial data.

### Funding

This work was supported by the National Social Science Foundation of China grant "Research on the Digital-Intelligence Transformation Mechanism and Path of the New Generation of Information Technology-Driven Financial Shared Services" (project number 21BGL040), the Soft Science Research Project of Henan Science and Technology Department Research on Information System Framework Based on Budget Performance Management (project number 202400410419), and the Philosophy and Social Science Planning of Henan Province Research on the dynamic mechanism and path of Henan's new urbanization development under the background of "three new and one high" (project number 2022BJJ119). The funders had no role in study design, data collection and analysis, decision to publish, or preparation of the manuscript.

### Grant Disclosures

The following grant information was disclosed by the authors:
National Social Science Foundation of China:  21BGL040.
Soft Science Research Project of Henan Science and Technology Department: 202400410419.
Philosophy and Social Science Planning of Henan Province:  2022BJJ119.

### Competing Interests

The authors declare there are no competing interests.

### Author Contributions

- Yuhua Chen conceived and designed the experiments, performed the experiments, analyzed the data, performed the computation work, prepared figures and/or tables, authored or reviewed drafts of the article, and approved the final draft.

- Hasri Mustafa conceived and designed the experiments, performed the experiments, analyzed the data, performed the computation work, prepared figures and/or tables, authored or reviewed drafts of the article, and approved the final draft.
- Xuandong Zhang conceived and designed the experiments, performed the experiments, analyzed the data, performed the computation work, prepared figures and/or tables, authored or reviewed drafts of the article, and approved the final draft.
- Jing Liu conceived and designed the experiments, performed the experiments, analyzed the data, performed the computation work, prepared figures and/or tables, authored or reviewed drafts of the article, and approved the final draft.

## Data Availability

The code is available in the Supplementary Files and data is available at Baidu AI Studio: https://aistudio.baidu.com/aistudio/datasetdetail/101589.

## Supplemental Information

Supplemental information for this article can be found online at http://dx.doi.org/10.7717/peerj-cs.1231#supplemental-information.

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
