# Peer review of "Design and analysis of management platform based on financial big data"

_PeerJ Computer Science, doi:10.7717/peerj-cs.1231_

## Round 0.1 · original submission · Minor Revisions

The review of your submission to PeerJ Computer Science is complete.

Although we found that your paper has merit, it is not acceptable to publish in its present form. We invite you to revise your paper to address reviewers’ comments as fully as possible and resubmit.

Reviewer 1 ·

Basic reporting

1- This paper uses a fuzzy clustering algorithm to classify financial data automatically and uses the LOF algorithm, which adds neighborhood relations to detect abnormal data. In addition, a financial data management platform based on distributed Hadoop architecture is designed. MapReduce framework is combined with fuzzy clustering algorithm and LOF algorithm, and MapReduce is used for parallel operation with the two algorithms to improve the algorithm's performance and accuracy. It is helpful to improve the operating efficiency of financial data processing.
2- The author needs to distinguish the purpose and significance of the research. At present, the research purpose of this paper is still not outstanding. Why do we need to optimize and classify financial data? This point needs to be emphasized in the introduction.
3- In the literature review (Section 2.2), the author lacks a summary of MapReduce improvement methods, which makes the innovation point of this paper vague.

Experimental design

4- The description of the MapReduce framework structure in FIG. 1 is not detailed enough. It is hoped that the author will add the working mode of corresponding modules, such as the data detection module, to form a good echo with the following article
5- As shown in Figure 2, the running state of a virtual machine is saved when it is running, so what is the meaning of "distributed storage service"?
6- Equation (5), as a linear discriminant function of classification statistics, seems to have no explanation for its determination method and construction basis, which makes me very confused.

Validity of the findings

7- After adding the neighborhood relationship, the LOF algorithm also considers the positive and negative k neighborhood distance to represent the density characteristics of data center points, but its quantitative relationship is still not clear according to Equations (12)-(15).
8- Comparing the running efficiency of NLOF algorithm on single machine and on cluster, the author does not discuss in depth.

Reviewer 2 ·

Basic reporting

Based on the distributed Hadoop architecture, this paper uses MapReduce, fuzzy clustering algorithm, and NLOF for parallel operation, which helps to improve the accuracy of the algorithm in the calculation of outlier factors and effectively avoids the probability of misjudgment on the edge of elements when multiple elements are too close to each other. In addition, this paper combines the business process to classify and identify the attributes of financial data, takes the fuzzy clustering distribution ofthee businessocess as the center vector, adopts the piecewise detection method to realize the automatic classification of financial data, and realizes the improved design of the classification algorithm. Using parallel algorithms to analyze big financial data can effectively improve the speed and accuracy of algorithm recognition; in my opinion, this is an exciting study, but the feedback of the author is not very good in the following issues, so I suggest accepting this paper after modification.

Experimental design

• In my opinion, Hadoop has good capacity scaling, and at the same time, it has good performance scaling and reliability. Therefore, the authors need to clarify the necessity of improving it;

• The author lacks necessary summaries in the literature review, including the improvingsting me, etc. I suggest reorganizing this part of the language;

• In Section 4.1.2, what is the basis for the construction of a linear discriminant function?

• I suggest that the author further introduce the method adopted. The traditional LOF algorithm calculates the outlier factor of each element in the data set then determines whether the element points are outliers.

• In the LOF algorithm with the addition of neighborhood relation, what is the intrinsic relation between the judgment basis of abnormal data points and P defined by the author? From the current description, this point needs to be discussed in depth;

• In the data analysis section, I would like to see the author analyze the effectiveness of the algorithm at different levels;

Validity of the findings

• Finally, the author needs to elaborate the practical significance and application scenarios of this research. Moreover, the authors must have their work reviewed by a proper translation/reviewing service before submission; only then can a proper review be performed. Most sentences contain grammatical and/or spelling mistakes or are not complete sentences.

---

## Round 0.2 · accepted · Accept

Thanks for your fine contribution to our esteemed journal.

Reviewer 1 ·

Basic reporting

I have reviewed the paper, all the commments are incorporated nicely. Now, the paper is in well manner and can be accepted to be published.

Experimental design

I have reviewed the paper, all the commments are incorporated nicely. Now, the paper is in well manner and can be accepted to be published.

Validity of the findings

I have reviewed the paper, all the commments are incorporated nicely. Now, the paper is in well manner and can be accepted to be published.